# LocSys: A Low-Code Paradigm for the Development of Cyber-Physical Applications

**DOI:** 10.3390/s25133951

**Published:** 2025-06-25

**Authors:** Konstantinos Panayiotou, Emmanouil Tsardoulias, Andreas L. Symeonidis

**Affiliations:** Faculty of Enginnering, Aristotle University of Thessaloniki, 541 24 Thessaloniki, Greece; klpanagi@ece.auth.gr (K.P.); symeonid@ece.auth.gr (A.L.S.)

**Keywords:** smart environments, cyber-physical systems, model-driven development, domain-specific languages, low-code development

## Abstract

Application development for the cyber-physical systems (CPS) domain is considered a quite complex procedure, since it not only requires a high level of expertise but also deep knowledge of heterogeneous domains. On the other hand, modern low-code solutions and DSLs are developed to offload domain complexity by developing models at a higher level of abstraction. In this work we propose an approach based on multiple high-level domain-specific languages (DSLs) as the vehicle to alleviate the developers from the intricacies of the CPS domain, enabling them to easily design and develop different layers (e.g., device, system or application layers) and aspects (e.g., automation processes, observation or monitoring dashboards) of a CPS. The materialized outcome of our approach is the LocSys platform, which allows the integration of DSLs, the development and management of models, and the development of pipelines of transformations between DSL models in a uniform platform, covering different aspects of complex domains. The efficacy of this approach was evaluated during a workshop that included more than 80 participants, with varying levels of expertise and experience in the field. The workshop documented the usability and acceptance of the study using SUS measurements. Preliminary findings suggest that the multi-DSL approach is highly usable (average SUS score 80.65, A− grade) and has been well received by non-domain experts. These results are promising, as they indicate that the LocSys platform can be successfully implemented to build smart environments with embedded automation processes and monitoring dashboards.

## 1. Introduction

The Internet of Things (IoT) envisions a world where everyday physical objects, such as sensors, robots, computing units, and smart bulbs/relays/switches/robots, among others, are connected to the Internet [1], acting as information sources, processing units, augmentation sources, and physical actuators capable of changing the state of the environment. These entities are commonly known as “smart objects”, possessing advanced computing, communication, sensing/activation, and storage capabilities, serving as fundamental components for IoT application development. On the other hand, cyber-physical systems (CPS) integrate sensing, computation, control, and networking into physical objects and infrastructure, connecting them to the Internet and to each other.

Developing applications for modern CPS remains a complex process, requiring expertise in various areas, including embedded systems, software development, computer and software engineering, communication protocols, and networking. Therefore, the effort required to develop, deploy, and maintain applications remains high. In addition, the domain’s high heterogeneity and complexity often lead to bugs during the development phase, which can critically impact the system’s smooth operation. To reduce the required effort and increase productivity while reducing errors, research and engineering efforts have focused on the development and adoption of low-code/no-code solutions [2,3,4]. These solutions are presented as platform and software frameworks that integrate domain know-how, including concepts, relationships, and rules. They offer the capability to automate software development processes and documentation elements, transforming a description model into software that is ready to install and run [5]. The implementation of these solutions is typically based on domain-specific languages (DSLs) [6] and follows the principles of model-driven engineering (MDE) and model-driven development (MDD) [7,8,9].

The current work focuses on the development of CPS applications, in the context of infrastructure digitization and smart environments, from citizen developers [10], an end-user category with minimal or zero technical and technological background, via a multi-lingual (DSLs) methodology that follows the decomposed meta-modeling (MM) approach. The rise of citizen developers utilizing low-code/no-code (LC/NC) platforms marks a transformative shift in the software development landscape. On the other hand, smart environments enable the acquisition and application of knowledge to improve the experience of their inhabitants [11]. Examples of heterogeneous smart environments include smart homes, offices, hospitals, campuses, and smart cities. These environments have different requirements and goals, but they all share the need for the IoT as a key enabling technology [12]. The proposed multi-DSL approach has been realized in the form a platform that enables the development of domain-agnostic, web-enabled, integrated development environments based on model-driven principles and domain-specific languages (DSLs), named LocSys. It allows the integration of independent DSLs via REST APIs, the execution of model-to-model and model-to-text transformations (e.g code generators) into ready-to-deploy applications, and the definition of transformation pipelines for complex multi-lingual domains, which is the case of cyber-physical systems and smart environments.

The overarching goal of the proposed methodology is to simplify the development process of complex systems for non-experts, such as the case of citizen developers, while reducing complexity and increasing end-user productivity. In the context of the current work, we evaluated our methodology against three research questions.

RQ-1: Can we accelerate the development process of applications for smart environments?RQ-2: Does our approach need minimum domain knowledge and is it framework-, and platform-agnostic?RQ-3: Can non-experts in the field of digital infrastructures and smart environments, such as citizen developers, use our approach?

The remainder of the paper is structured as follows. Section 2 presents the latest advances and related work in the context of MDD and DSL focused on application development for the IoT and CPS domains. Section 3 discusses in detail our multilingual development of cyber-physical systems, following a decomposed meta-modeling approach, while Section 4 describes the implementation of this approach via the LocSys low-code platform as well as the DSLs created to evaluate the approach. Next, Section 4 focuses on the empirical evaluation conducted in the context of this study to assess the effectiveness and productivity gain of non-experts in developing applications for smart environments. It also includes an evaluation of the approach based on metrics and discusses the results, limitations, and threats to the validity of our work. Section 5 summarizes our conclusions and discusses future work before concluding the article.

## 2. State of the Art

The landscape of low-code development is undergoing a period of rapid evolution, particularly in the context of DSLs for CPSs and smart environments. This evolution is driven by the increasing demand for efficient software development processes that can accommodate the complexities of modern applications, particularly those that integrate physical and digital components.

Low-code development platforms (LCDPs) are gaining traction as they facilitate the creation of applications by users with minimal programming skills, often referred to as “citizen developers”. These users can employ visual interfaces and pre-built components to develop software through a process that is increasingly being described as the “democratization” of software development. This is particularly beneficial in the context of CPS and smart environments, where rapid prototyping and iterative development are essential to adapt to changing requirements and technologies [3,13]. The use of DSLs within these platforms enables the provision of these solutions that are tailored to the specific requirements of a variety of domains, thus enhancing both the usability and functionality of the platforms in question [14].

The importance of MDE principles in low-code development is increasingly being recognized in the academic literature. MDE provides a means of abstracting complex systems, enabling developers to concentrate on the high-level design of a project rather than being constrained by the necessity of low-level coding [5,15]. This is particularly pertinent to CPS, where systems often involve complex interactions between hardware and software. DSLs are low-footprint, declarative languages that enable developers to write code at the appropriate level of abstraction using concepts from the underlying domain. They provide an effective means for the domain experts to specify solutions and can be regarded as a specialization of MDE. Using DSLs, developers can construct models that accurately reflect the behavior and requirements of these systems, thus facilitating improved communication between stakeholders and more efficient development processes [16].

Furthermore, the incorporation of low-code platforms with IoT technologies represents a notable trend. As the number of IoT devices continues to grow, the need for platforms that can effectively integrate these devices into larger systems becomes increasingly apparent. Low-code solutions are increasingly designed to support IoT applications, thereby facilitating the rapid deployment and management of interconnected devices [2]. This trend is reinforced by the emergence of event-driven architectures, which are particularly well suited to the dynamic nature of smart environments [17].

The performance and scalability of low-code applications are also being subjected to rigorous examination by organizations seeking to ascertain their ability to meet the demands of enterprise-level applications. Research findings suggest that, while low-code platforms can facilitate accelerated development, they must also address challenges related to technical debt and maintainability. It is imperative that low-code applications are subjected to continuous monitoring and evaluation to mitigate the aforementioned risks and ensure their long-term viability [18].

However, the utilization of DSLs in the context of modern CPS and Smart Environments is becoming increasingly prevalent [12,19], largely due to their capacity to streamline the design, implementation, monitoring, and remote control processes [20]. These languages are tailored to specific application domains, allowing developers to construct systems with minimal complexity and enhanced efficiency. Next, we present a selection of notable studies on low-code development and DSLs, which mainly focus on the application layer.

The field of research pertaining to the Internet of Things (IoT) and modern cyber-physical systems (with networking and integration capabilities with IoT, edge, and cloud platforms and infrastructures) is extensive, encompassing a multitude of disciplines. These include, but are not limited to, networking and communication protocols, software and hardware engineering, and data analytics. Conversely, research in other areas, such as the automation of software development, implementation and deployment processes, scalability, reliability, and verification, remains comparatively limited [21]. In [2], the authors discuss the state of the art on existing approaches that support the development of IoT systems and focus on DSLs and tools available in the MDE domain and on emerging low-code development platforms covering various aspects of the IoT domain. The results present all the characteristics of a typical modeling platform supporting IoT system development and various limitations for each solution have been identified and documented in the qualitative analysis published. In addition, an interesting systematic review and comparative analysis for DSLs intended for the IoT and CPS domains was recently published in [21]. In their paper, Sadik Arslan et al. conducted an in-depth analysis of 32 distinct domain-specific languages (DSLs) designed for software development. These were categorized based on a set of defined requirements, resulting in three distinct groups: (a) language definition, (b) language features, and (c) tool support. Furthermore, an analysis of the non-functional characteristics of DSLs was conducted, including performance, security, data protection, reliability, availability, programmability, scalability, and resource consumption. We claim that the findings of this study will prove invaluable in guiding the usage and deployment of DSLs in the IoT and modern CPS domains. Furthermore, they will assist in selecting the most appropriate DSL for a given deployment scenario, whether at the device, system, or application level. In addition, their study will help to identify and categorize other studies and tools related to the development and use of MDD tools, DSLs, and low-code platforms, thereby reducing the complexity of system and application development.

An intriguing discrete-time signal processing approach for modeling the behavior of sensors in wireless sensor network (WSN) architectures is FRASAD, which was first presented in 2015 by Xuan Thang Nguyen and colleagues [22]. The objective of FRASAD is to enhance the reusability, flexibility, and maintainability of software deployed on sensor devices in a WSN. A software architecture and a rule-based programming model are proposed, which allow the description of the behavior of sensor nodes using domain concepts. One of the novelties of this approach is the possibility of converting the models into executable software for the operating systems TinyOS [23] and Contiki [24]. The platform-independent model (PIM) is initially mapped to the intermediate program code using C language constructs and functions. In the subsequent translation phase, the process generates the operating system-specific application files from the intermediate program code, taking into account the operating system layer. This results in platform-specific executable code. It should be noted, however, that the FRASAD DSL is limited to WSNs and the behavior of sensor devices based on a sense–plan–act model. The latest developments in the field of Internet of Things (IoT) technologies, such as the emergence of asynchronous and bidirectional communication protocols and message brokers (e.g., MQTT, Kafka and AMQP brokers), have led to a need for a more abstract approach to the development of applications in this domain. This is particularly relevant when considering the integration of already connected devices, such as sensors, actuators, and robots. Modern DSLs must, therefore, be capable of integrating and interacting with a diverse range of IoT devices, while also enabling the development of applications that are entirely independent of the underlying low-level hardware complexity.

Conversely, Einarsson et al. put forth the proposition of SmartHomeML [25], a DSL for the design of applications for smart homes. SmartHomeML provides a model design environment and the capacity to utilize M2T transformations to automatically generate mesoscale connectivity for smart home devices that comply with the specifications of a selected control system. The meta-model language delineates the interaction of a “smart thing” with cloud computing. Despite its intriguing nature, the approach is constrained by the fact that it only supports interactions over the internet, which introduces limitations in terms of its applicability to more complex systems.

Another noteworthy study is IoTDSL [26], a DSL for defining possible actions of an IoT device that must respond to real hardware platform events. This DSL is accompanied by a grammar for defining rules in the execution of actions based on certain conditions. However, IoTDSL assumes predefined communication paths and there is no possibility to create source code to automate the software development process. Furthermore, the contributions of CyprIoT [27] and ThingML [28,29,30,31] are worthy of note. The former provides tools for developing IoT applications and includes two distinct languages: one for modeling the network and one for verifying its correctness through predefined rules. Conversely, ThingML is a language employed for the purpose of designing the manner in which information is to be received from peripheral devices. It is utilized by third-party DSLs (e.g., CyprIoT) for the purposes of modeling and automatic software generation for the specific platform. Despite the considerable advantages presented by this approach, it is deficient in a comprehensive approach dedicated to modeling and deployment at the application layer, where devices and services are considered to be already connected (to a broker or platform at the Edge or in the Cloud).

One interesting work on DSLs in the field is presented in [32], which focuses on modeling the network and communication layer of an IoT system. The authors introduce a networking-oriented methodology to unify heterogeneous concepts in the domain and extend the CyprIoT and ThingML languages to enhance the model with network-related properties. They also avoid the inherent heterogeneity of the IoT domain by separating the network specification (the physical entities, communication layer, and constraints) from its concrete implementation (the source code and documentation). Furthermore, to address the need for future IoT systems to integrate heterogeneous components and manage weak connectivity, this paper introduces ComPOS [33], a DSL designed for composing IoT services. Although a small language, ComPOS offers robust message mediation capabilities through stateful reactions that support nested and parallel message sequences and anonymous futures. The authors also note that alternative strategies to this abort semantic can be implemented by incorporating strategy services. The practical viability of ComPOS and its approach to weak connectivity was evaluated through the implementation of seven distinct home automation scenarios.

To address the challenges in engineering complex and heterogeneous IoT systems, the authors in [34] propose IoTMoF, a model-driven framework that facilitates requirements development, design, and code generation by employing platform-independent modeling (PIM) through its ReqMIoT environment. This environment supports use-case modeling, notably, incorporating the specification of exceptional behaviors and their handling, and generates IoT ARM compliant domain models. These PIMs are then transformed into platform-specific models (PSMs), specifically, an IoT information model and a state-chart model that details both normal and exceptional system operations. Finally, IoTMoF uses these models, along with a device configuration model, to automatically generate code, including wrapper code for deployment on the target IoT platform, as demonstrated with a smart lights system example.

The aforementioned research effectively addresses various, mostly low-level, aspects of the field of IoT and modern CPS. However, none of the studies focus on the application layer and the process of developing applications for smart environments. Currently, smart objects are connected to middleware and platforms and users can directly participate in the development of the application logic layer, which is the case in the current study.

In conclusion, the latest trends in low-code development using DSLs for complex systems and smart environments reflect a broader shift towards more accessible, efficient, and adaptable software development practices. By leveraging the strengths of low-code platforms and DSLs, organizations and practitioners can better meet the challenges posed by complex, interconnected systems while fostering innovation and collaboration among diverse teams.

## 3. Methodology

In the context of modern CPS, devices (sensors, actuators, processing and storage units, etc.) and robots are connected and allow remote access to their resources. The CPS domain is inherently interdisciplinary and integrates concepts and technologies from various other fields, such as robotics, distributed systems, hardware and software engineering, and embedded systems, among others, making their development a complex process that is usually carried out in teams composed of experts in these different fields.

### 3.1. CPS Domain Decomposition

In order to effectively address the inherent complexity of CPS, we propose a decomposition of the problem into sub-domains, organized into three distinct layers and intermediate ones, as illustrated in the diagram presented in Figure 1. In particular, the lowest layer (L1) is concerned with the development of devices that manage peripheral sensors/actuators and/or robotic systems equipped with sensors/actuators, which provide access to higher-level functions. The subsequent layer pertains to the development and definition of the system (L2), while the third level encompasses the development of applications and processes (L3). However, there are two intermediate layers between L1–L2 and L2–L3. These layers facilitate communication between devices and applications and the system layer, which are collectively referred to as ILC (intermediate layer communication). It is important to note that ILC 1–2 and ILC 2–3 may be independent of each other, with the potential to provide different interfaces and communication protocols. Of course, L1 and L2 levels can be further separated into sub-domains (layered), which can be defined in terms of the functionality provided and the design and development stages involved. The rest of the sub-section will examine these individual layers in greater detail.

#### 3.1.1. Device Layer (L1)

The device layer, also known as the physical layer, is a fundamental layer responsible for the management and control of the physical entities of the system. It encompasses a wide range of devices, including sensors, actuators, embedded controllers, robots, and other physical elements that interact with the physical environment. This layer pertains to the development process of IoT and embedded devices. It is, in fact, further divided into two principal sub-phases: (a) the design and development of the device/robot’s hardware, which encompasses the processing unit, peripherals, power management, wire connections, and protocols such as SPI, I2C, and UART, and (b) the development of the software to control the device and the connected peripherals, as well as to connect and communicate with remote edge/cloud machines and platforms, which constitute the firmware of the device. However, an intermediate level could be defined, that of hardware–software, which concerns the development of the hardware/peripheral drivers and practically links the two levels.

The development of device-level software to control peripheral sensors and actuators is one of the most difficult and time-consuming processes in the IoT and CPS domain, due to the necessity for individuals engaged in this process to possess specialized training in low-level hardware/firmware-related concepts, technologies, and tools. Furthermore, the multitude and heterogeneity of available devices (processing units, development platforms, and peripheral sensors and actuators) and networking, communication, and messaging protocols [35,36] (such as REST-HTTP, Zigbee, LoRa, Bluetooth, BLE, CoAP, and MQTT) add high complexity and complication to the design, development, implementation, and deployment processes.

#### 3.1.2. Communication Layer (LIC 1-2, LIC 2-3)

The networking and communication layer, as well as the selection of appropriate technologies for its deployment, has always been an important factor in the abstraction of communication and application development in CPS. This layer is responsible for managing the flow of data, ensuring real-time or near-real-time communication, and providing the infrastructure with the interaction between physical and digital resources in a CPS. The networking and communication layer plays a critical role in the exchange of data and information between the various terminal nodes in the system, including sensors, actuators, robotic systems, controllers, and high-level applications.

The most common approach to implement communication patterns in the CPS domain is through message broker technologies, which satisfy the above requirements, in terms of their characteristics and the functions they provide in the context of CPS. Several such technologies exist, such as Redis (https://redis.io/, accessed on 1 June 2025), EMQX (https://www.emqx.com/en, accessed on 1 June 2025), RabbitMQ (https://www.rabbitmq.com/, accessed on 1 June 2025), and Apache Kafka (https://kafka.apache.org/, accessed on 1 June 2025), each optimized for specific functionality within the CPS ecosystem. These mediators support a number of communication protocols, such as AMQP, MQTT, CoAP, and STOMP, and they also provide appropriate mechanisms for communicating over the Internet, i.e., using HTTP and Websocket protocols.

#### 3.1.3. System Layer (L2)

In the context of CPS, the system layer is one of the key architectural layers, playing a critical role in the overall operation and coordination of the system. This layer is usually located between the physical layer (device) and the application layer in the architecture of the CPS and serves as a bridge between the physical world and the software applications that control and monitor the physical processes. The system layer provides the appropriate tools for easy interaction between the physical and digital worlds, including the integration and coordination of CPS entities, data aggregation, real-time processing, management and control of assets, communication management, resource distribution, fault tolerance, security, and overall monitoring.

#### 3.1.4. Application Layer (L3)

Finally, the application layer is another important layer in the architecture of a CPS, mainly concerning the development of process automation applications and user interfaces to systems, such as graphical environments (e.g., dashboards) for remote monitoring and control. At this level, high-level functionality and system logic are implemented for specific domains. The application layer is responsible for the orchestration and management of the overall operation in a CPS to achieve specific goals and objectives, including a higher level of control and logic, task management, user interfaces, performance monitoring, fault management, and recovery, as well as interaction with physical entities (sensors, effectors, robots, etc.).

### 3.2. The Decomposed Multi-DSL Approach

From the aforementioned descriptions, it is evident that complex domains such as CPS and smart environments are composed of multiple layers that exhibit varying degrees of technical correlation, whether minor or significant. For instance, the device layer and the communication layer may be developed by a single technical individual, given that the configuration of a device frequently necessitates the establishment of connectivity. However, it is challenging for a single developer to simultaneously configure a device and implement a dashboard, or to allocate tasks among the CPS entities at the application layer.

It is, therefore, recommended that complex domains, such as CPS, should not be meta-modeled as a whole, but rather decomposed into different DSLs for each layer, with one for each sub-domain. This approach allows for each layer (or part thereof) to be modeled by experts, thus alleviating the inherent technical complexity for end users, as evident in Figure 2. In order to achieve this, the abstract and concrete meta-modeling process must adhere to the structured programming or the OOP (object-oriented programming) paradigms. This allows for the development of several components that can then be imported for utilization into a new piece of software, which, in the case of DSLs, refers to grammar modularization, thereby streamlining the development process. It is, therefore, necessary to define the core abstract and concrete (grammar) meta-models that can be composed into more complex ones, thus providing the formalization of layer-specific domain-specific languages (DSLs). Examples of core grammars may include Entity grammar, DataType grammar, Condition grammar, and so forth, which can be utilized in more than one CPS layer. In this way, the CPS domain meta-modeler is equipped with a number of fundamental components, which can be readily employed to define or extend existing models in order to create new concrete meta-models for all involved layers.

The benefit of the decomposition of the CPS domain at the meta-modeling layer is that several DSLs that handle different aspects of the same CPS may exist, models of which are created either by experts of the sub-domain or by citizen developers (Figure 2). The existence of multiple domain-specific languages (DSLs) for a single complex domain has significant implications for the creation of DSL pipelines. These pipelines can be conceptualized as semantic graphs, comprising DSL models as nodes and DSL-related operators as edges. The most prevalent operator within this context is the M2M (model-to-model) transformation.

In this way, CPS development may be initiated by building models in device-related DSLs, which are then transformed into system-related DSLs and enhanced with additional domain-specific knowledge. Finally, the models are transformed into application-layer DSLs, where the end user can define the business logic and the behavioral aspects of CPS applications.

## 4. Implementation

In order to materialize the domain decomposition and multi-DSL methodology, a low-code web-based platform named LocSys was materialized. LocSys was designed to host heterogeneous DSLs in type (graphical/textual) and domain, allowing the development and management of DSL models. In order to support model operators, LocSys provides a common interface for the integrated DSLs, offering standardized operators that can be applied in all models via a REST API.

### 4.1. DSL Model Operators

LocSys implements five model operators, namely, validation, import, transformation, generation, and deployment. These comprise the toolset to create model pipelines, since they can interconnect models of different DSLs and produce deployable code. This is considered important in complex domains, in order to transform models between DSLs covering different aspects or levels of the domain. For example, from a DSL model that describes a smart environment, a pipeline of operators can be defined to transform to DSL models that can describe an automation process or the monitoring dashboard of the environment. The definition of each operator is provided below.

The validation operator is applicable to all DSLs and is one of the most basic MDE operations, since when applying the validation operator to a model, its correctness is checked against the MM of the specific language, but also with respect to its syntax. The successful validation of a given model is a prerequisite for the subsequent application of the transformation, code generation, and code execution operators.

The import operator concerns the integration of a DSL model into another DSL model or related DSL. Imports are often necessary since some models (hence, some DSLs) must necessarily include information from a previous model in order to make sense or in order to be able to perform proper validation. This process is similar to the process of import in programming languages, which may involve incorporating functionality from third-party libraries/modules and already existing source code (e.g., reusing functions, classes, etc.). Thus, the use of import operations promotes the reusability of commonly used concepts across DSLs, such as data types, messages, connectivity and communication models, device configurations, and others. This operator is not necessarily applicable to all DSLs.

The transformation operator refers to the second fundamental operation of the MDE methodology, in which the model in question is transformed into another model of another conceptually related DSL. This type of transformation can be observed as a language transpiler [37]. The way in which models are transformed varies, and the most common involves the use of transformation rules that specify how concepts/rules of the input model are mapped into the output model. This operator need not be present for all DSLs and is allowed to be executed after the model has been properly validated.

The code generation operator converts the model in question into executable code, which may consist of one or more files. This operator is essentially a version of the transformation operator since here, too, a transformation is performed, not to another DSL, but to the code of a GPL (general purpose language). It can be argued that, in the context of a code segment such as Python, the concept of a model of the Python language can be considered to be a formal transformation, given that it adheres to the established rules and syntax of the language. Nevertheless, LocSys adopts a conceptual separation of transformation and generation. This is based on the assumption that the result of the transformation is a model of a DSL hosted on the LocSys platform, which can then be subjected to further processing through the usage of operators. In contrast, the result of the generation is executable code. This operator need not be present for all DSLs and is allowed to be executed after the model has been successfully validated.

The ultimate objective of the LocSys platform is to eliminate any technical burden on the end user wherever possible. Consequently, for the DSLs for which it is appropriate to execute the generated code, an automatic execution mechanism has been incorporated. This way, if a model is successfully validated and supports code generation, the end user will be able to successfully execute the code, with minimal effort and possibly without relevant technical knowledge. This is the deployment operator, which needs not be present for all DSLs and is allowed to be executed after the model has been successfully validated.

### 4.2. Smart Environment Pipeline

As discussed in Section 3.2, a pipeline is essentially a flow-oriented description of successive model operators that produces several outcomes (other models, code, or deployments). In order to validate the decomposed multi-DSL approach, we developed four DSLs (two graphical and two textual), which are interconnected via the transformation and import operators, implementing a pipeline through which a user can define an environment, their smart devices, smart automations for them, and visualization components for inspection purposes. The implemented pipeline is presented in Figure 3.

The initiating DSL is EnvMaker, a graphical language to define the blueprint of the environment, followed by EnvPop, a graphical DSL via which sensors and effectors are declared. The third DSL is SmAuto, through which automations can be created using the entities defined in EnvPop, and lastly, CodinTxt visualizes sensory data and provides interfaces for actuator commands. Referring to the decomposed MM approach, EnvMaker and EnvPop belong to the system layer, as they define system aspects of the CPS, whereas SmAuto and CodinTxt are part of the application layer, since they provide logic and inspection. It should be stated that no communication-layer DSL was created, as it is being handled inside the EnvPop DSL, where the communication means for each entity is defined, propagated to the SmAuto and CodinTxt via M2M transformations. Finally, the EnvPop entities are mock (i.e., they are not correlated to physical devices, but produce user-defined value distributions); therefore, no DSL of the device layer was used. This was intentional, since the device layer is notoriously difficult to master, a fact that makes it far from ideal to utilize in the verification of our methodology by people with low/medium experience in programming and/or CPS. Furthermore, due to the decomposition of the domain complexity into layers with common communication middleware (message broker) between layers (see Figure 2), our approach is device-agnostic, and the application layer considers already connected physical and virtual entities (such as the case of device-related sensory and actuation entities) to a message broker. Next, the involved DSLs will be described in depth.

#### 4.2.1. EnvMaker DSL

The first DSL of the implemented pipeline is EnvMaker, a graphical DSL that allows for blueprint creation. Each blueprint is defined as a collection of tiles, placed in a grid with user-defined number of rows and columns. EnvMaker supports tiles from five different domains, namely Home, Blueprint, City, Farm, and Supermarket, also allowing for placing tiles from different domains in the same grid. The model creator can drag and drop tiles from a sidebar into the grid and rotate them by clicking on them. An example of a 3 × 3 grid resembling a house environment is visible in Figure 4. Since each graphical DSL has an internal textual representation (usually stored as the instantiation of a data model), EnvMaker stores each model as a JSON object, which contains information about the number of rows and columns as well as each tile (the row and column it belongs to, which tile it is, and its rotation). EnvMaker DSL does not support manual validation (explicitly from the user) since the user cannot create an erroneous model by design, nor any of the other model operators.

#### 4.2.2. EnvPop DSL

The second DSL in the CPS pipeline is named EnvPop, as it populates the environments created with EnvMaker with smart sensors, effectors, obstacles, and environmental properties (e.g., a source of water or a fire). EnvPop is also a graphical DSL, since it allows the user to drag and drop any device into the environment’s blueprint and change its properties by clicking on it. Prior to this, the user is obligated to import an EnvMaker model into the current EnvPop model, since the EnvPop items must be placed inside a defined environment (blueprint).

Regarding sensors, EnvMaker supports ambient light, distance, gas, temperature, humidity, and pH sensors, as well as two types of alarms, namely, area and linear alarm (an area alarm is triggered when an agent enters a defined radius near the alarm, and a linear alarm is triggered when an agent crosses the line between its two parts). When actuators are concerned, EnvPop offers humidifiers, lights, speakers, switches, and thermostats. Each one of these devices supports three modes of operations, real, simulation, or mock, where real assumes that a real device exists and the user is creating a digital twin, simulation means that the behavior of the device will be governed by a simulator, and mock means that sensors will dispatch mock values that follow specific distributions (constant, random, triangle, normal, and sinus). In Figure 5, the imported EnvMaker model is seen, populated with several sensors and effectors. Additionally, a temperature sensor has been selected, and its corresponding properties have been set to a dispatch rate of 3 Hz, mock mode, or operation. This has resulted in the generation of values that follow a sinus distribution.

Regarding the applicability of LocSys operators, similarly to EnvMaker, EnvPop does not support manual validation, as all models are correct by design. However, EnvPop requires the import of an existing EnvMaker model and supports the transformation to an SmAuto model, a DSL that will be described next.

#### 4.2.3. SmAuto DSL

SmAuto is the third step in the CPS pipeline, since it takes input from an EnvPop model and enables users to program complex automation scenarios, for IoT devices, that go beyond simple tasks [38]. SmAuto is a textual DSL, whose syntax was created using the textX DSL development framework [39].

The fundamental concepts of SmAuto are brokers, entities, automations, conditions, and actions. In essence, an SmAuto model comprises information regarding the diverse devices that comprise a smart environment (e.g., lighting, thermostats, smart refrigerators, etc.), the manner in which they communicate, and the automation tasks associated with them. The meta-model of SmAuto is illustrated in Figure 6. SmAuto is constituted by a series of “smaller” meta-models, including the BrokerMM, DataTypeMM, ConditionMM, and MessageMM, which provide the concepts of broker connection, attribute, and condition horizontally to LocSys DSLs.

In order to better explain the SmAuto concepts, a model example is provided below. In the model of Listing 1, an MQTT broker named home_broker has been defined, whose connectivity parameters and credentials have been declared. Each broker acts as the communication layer for messages where each device has its own topic, which basically resembles a mailbox for sending and receiving messages. SmAuto supports brokers which can serve the MQTT, AMQP and Redis protocols.

**Listing 1.** SmAuto model exampleBroker<MQTT> home_broker    host: "localhost"    port: 1883    auth:        username: " "        password: " "end

Entity weather_station
    type: sensor
    topic: "porch.weather_station"
    broker: home_broker
    freq: 1
    attributes:
        - temperature: float -> gaussian (10, 20, 5) with noise gaussian (1,1)
        - humidity: int -> constant (78)
end

Entity aircondition
    type: actuator
    topic: "bedroom.aircondition"
    broker: home_broker
    attributes:
        - temperature: float
        - mode: str
        - on: bool
end

Automation start_aircondition
    condition:
        (weather_station.temperature > 32) AND
        (aircondition.on is false)
    enabled: true
    continuous: false
    actions:
        - aircondition.temperature: 25.0
        - aircondition.mode:  "cool"
        - aircondition.on: true
end


Next, two entities follow, a weather_station and an aircondition Entities are connected smart devices that send and receive information using a message broker. For each entity, a name, a broker, a topic, and a set of attributes must be declared, where attributes define the data structure and the type of information dispatched through the defined topic. Furthermore, for each entity, its type should be defined, choosing a value from the sensor, actuator, and hybrid enumeration. The syntax for all types remains the same; nevertheless, a sensor is expected to publish information, whereas actuators consume information (take commands). *Sensors* have an additional field called freq, which defines the data publishing frequency. In our example, the weather_station sensor is connected to the home_broker and dispatches one message per second in the porch.weather_station topic, having as payload one attribute named temperature of type float and one attribute named humidity of type integer. Similarly for the actuator aircondition, the payload it expects to receive contains a temperature value, a mode string, and an on boolean flag. A special mention should be made for the weather_station attribute syntax, where it is showcased that the value generators defined in EnvPop are transformed into the produced SmAuto model.

The most prominent component of SmAuto is automation, which defines simple or complex rules that usually activate actuators based on conditions involving devices’ attributes. In the example of Listing 1, there exists a single automation called start_aircondition that changes the device’s mode to cool, turns it on, and sets the temperature at 25 degrees Celcius if the air condition is turned off and the temperature captured from the weather_station is more than 32 degrees. Of course, this is a simple automation, as SmAuto supports an FSM-like definition of automations. Specifically, each automation may have the following attributes:**condition**: The condition used to determine if actions should be run;**enabled**: A boolean flag defining if the automation can be executed or not;**continuous**: Whether the automation should automatically remain enabled once its actions have been executed;**checkOnce**: If true, the condition of the automation will run only once and then exit;**actions**: The actions that should be run once the condition is met;**after**: The automation will not start and will be held at the IDLE state until termination of the automations listed here as dependencies;**starts**: The automations that are enabled after termination of the current automation;**stops**: The automations that are disabled after termination of the current automation.

Using these properties, one can define an FSM-like flow of automations, much more complex than the ones supported by standard if-this-then-that tools. Regarding model operators, SmAuto facilitates the following processes: (a) validation, which entails the use of the TextX toolchain for syntactic verification of the current model, (b) generation, whereby the model undergoes transformation into Python code that is subsequently deployed, and (c) transformation, since the current model can be transformed into a CodinTxt model, via which a dashboard is automatically created towards system inspection. This enables the generation of a dashboard for the purpose of conducting a comprehensive examination of the system.

Finally, it should be stated that the Commlib library [40] is utilized in the generated code of SmAuto, implementing the ILC 2-3, i.e., the layer that connects the system layer to the application layer. Commlib is an internal DSL for communication and messaging in cyber-physical systems, and it can be used for rapid development of the communication layer on-device, at the edge and on the cloud. It implements a simple protocol-agnostic API (AMQP, Kafka, Redis, MQTT, etc.) for common communication patterns in the context of cyber-physical systems, using message broker technologies. Such patterns include PubSub, RPC, and preemptive services (aka actions), among others. Based on the above, Commlib can also be used as the implementation of a potential ILC 1–2 layer, since it abstracts several protocols.

#### 4.2.4. CodinTxt DSL

The final step of the CPS pipeline is CodinTxt, a textual DSL created to automate the development process of dashboards for cyber-physical systems. CodinTxt defines a meta-model for the Codin platform ( https://codin.issel.ee.auth.gr/, accessed on 1 June 2025), making it platform-specific, and allows the definition of dashboards using textual semantics, while it also provides a model-to-text (M2T) transformation for generating JSON descriptions of dashboards that can be imported into Codin. Its syntax was defined using TextX and its grammar is quite simple, as it comprises a metadata definition, broker definitions, and visual components. Regarding the concepts, metadata contains basic information concerning the model (author and description) as well as a token via which the automatic deployment of the model in the Codin platform is performed. The model author can declare several broker definitions, essentially, the communication channels to which the Codin platform will subscribe and visualize the data. Finally, several visual components exist (gauge, logsDisplay, valueDisplay, plot, buttons, JSONViewer, etc.) which are used as dashboard components that visualize values included in declared broker topics. The CodinTxt meta-model is shown in Figure 7.

An example of a CodinTxt model can be seen in Listing 2, where data from a humidity sensor are viewed in a JsonViewer dashboard component upon arrival.

**Listing 2.** CodinTxt model exampleMetadata    name: "lceoQ"    description: "BRIEF_DESCRIPTION"    author: "AUTHOR_NAME_HERE"    token: "..." // ** The Codin token **end

Broker<MQTT> locsys_broker
    host: "..."
    port: 8883
    ssl: True
    webPath: "/mqtt"
    webPort: 8894
    auth:
        username: "..."
        password: "..."
end

Entity sn_humidity_1
    type: sensor
    topic: "lceoQ.sensors.sn_humidity_1"
    broker: locsys_broker
    attributes:
        - humidity: float
end

// Transformed from sensor Entity sn_humidity_1
JsonViewer sn_humidity_1Display
    label: "sn_humidity_1 Display"
    entity: sn_humidity_1
    position:
        x: 0
        y: 0
        width: 4
        height: 4
end


CodinTxt supports (a) validation, since the corresponding model’s validity must be ensured against the TextX syntax/grammar, and (b) the deployment operator, which directly deploys the model as a dashboard in the Codin platform and returns its URL.

## 5. Evaluation

As already discussed, the proposed multi-DSL methodology aims to ease the development process of complex systems (in our case, smart environments) for non-experts, i.e., citizen developers, while lowering complexity and increasing the productivity of end users. The evaluation is triggered by the four research questions earlier stated in Section 1, focuses on usability, and is related to domain expertise productivity metrics.

### 5.1. Formulation

In the context of the current work, we conducted a two-hour midscale workshop to present and evaluate our low-code approach for developing applications for smart environments via the LocSys platform, where 84 users participated. Some were familiar with programming, a few with Python and DSLs, but most of them had limited programming knowledge and no knowledge in the domains of the IoT and CPS, an overall background profile similar to the average citizen developer.

In the first part, a very brief presentation (30 min) of the basic concepts of smart environments in the context of the CPS was provided, including the definition of cyber-physical systems, what low-code/no-code DSLs are, and the operators that can be applied to models (as described in Section 4.1). Then the LocSys platform was showcased live, and a basic example of the full pipeline was shown. Then, four tasks were given sequentially, one for each DSL of the CPS pipeline, allowing the participants to define their mock CPS, create and deploy automations, and create and deploy the related dashboards. The required tasks, along with the time provided to submit, follow. The total hands-on time of the participants, including the fifth task that was experimentation on their part, was close to an hour (55 min).

Task 1 (5 min): Create an environment of size 4 × 4 tiles, using EnvMakerTask 2 (10 min): Import this environment in an EnvPop model and add−a humidity sensor (mock, normal distribution, μ = 60, std = 10);−a temperature sensor (mock, random distribution, min = 18, max = 25);−a thermostat actuator (initial value 20°).Task 3 (30 min): Transform this to an SmAuto model and add two automations:−If the temperature < 19, set the thermostat to 23°;−If the temperature > 24 and humidity > 75%, set the thermostat to 21°.Then, validate the model, generate the merged code, and execute it in a Google Collaboratory notebook.Task 4 (10 min): Transform the SmAuto model to CodinTxt and add your Codin token. Then, deploy the dashboard!Task 5 (no time): Go back and experiment with sensors, actuators, automations, and dashboard components!

After concluding the workshop, a questionnaire was electronically provided to each participant to report their experience and give feedback on our approach and the multi-lingual development methodology and pipeline. The questionnaire adopted the System Usability Scale (SUS) [41,42] empirical evaluation approach, which is a highly robust and versatile tool for usability practitioners to quickly and easily assess the usability of a given product or service [43].

### 5.2. Results

In this section, the results of the performed experiment are described. Initially, information regarding the participants’ experience in related domains is described, analyzing their responses to the aforementioned questionnaire. Then, a commentary on the DSL utilization will be performed, showcasing the participants’ success rates in executing the specified tasks. Finally, the overall methodology and the LocSys system are evaluated against their usability by analyzing the SUS scores gathered from the questionnaires.

Regarding the user experience in the involved domains, a Likert scale from 1 to 5 was used, where 1 stands for “I know nothing of the domain” and 5 stands for “I am an expert in the domain”, as well as true/false questions. Of the 84 participants, 45 of them answered the questionnaire, the results of which are visible in Figure 8. Regarding programming experience, the results showed that most of the participants had medium to high experience (41.9% reported a programming experience of 3, and 74.5% of 3 or 4 combined), having a mean value of 3.23 (Figure 8a). Furthermore, Figure 8b indicates that almost two out of three had never used a domain-specific language for creating software components or snippets. Regarding familiarization to Cyber-Physical Systems, the participants’ experience can be described as low, with a mean value of 2.16, since 72.1% of the participants provided a score of 1 or 2, while none of them considered themselves experts (Figure 8c). Furthermore, 86.7% and 82.2% stated that they had never been involved in any way in the development of a CPS (Figure 8d) or in the development of a CPS application, respectively (Figure 8e). Finally, concerning Smart Home Automations, the experience of the participants can be described as low/medium, with a mean value of 2.49, since 81.4% responded with 2 or 3, while none of them considered themselves experts (Figure 8f). Specifically, 71.1% and 73.3% stated that they had not participated in any way in the development of smart home systems and applications/automations (Figure 8g,h). Consequently, we can argue the participants had limited domain knowledge and expertise and their SUS results can be utilized to answer RQ-2 and RQ-3.

Regarding the actual implementation part, all 84 participants fulfilled the required first four tasks, creating 84 models for each DSL (EnvMaker, EnvPop, SmAuto, and CodinTxt), resulting in 344 total models and 943 total model validations in an hour. It should be stated that within the 4-DSLs pipeline, participants actively used the validation operator only in the SmAuto case, as this is the only part that required manual textual input that would define the automations. The prior is a fact since EnvMaker and EnvPop are graphical DSLs and, thus, the models are correct by design, and CodinTxt models (although textual) were generated after direct transformation from valid SmAuto models; therefore, they were valid by design as well. In our experiment, only 3 out of the 84 participants did not manage to produce a valid SmAuto model that adheres to the task requirements in the time range of 30 min (Task 3); the participants managed to create correct and valid automations at a rate of 96.42%. If we consider all the models that participate in the pipeline, this means that the valid generated models of the whole pipeline were 333 out of 336 (336 is computed by multiplying 84 models by 4 DSLs), achieving an astounding success rate of 99.11% in generated models. These results indicate that with a success rate of 96.42%, participants that had low to medium experience in smart home automations and CPS managed to produce a working environment that (a) generates values, (b) executes automations, and (c) visualizes its status in a dashboard, in less than an hour, a fact that showcases the strength and potentials of the distributed MM methodology and the DSLs pipelines as well as the incorporation of mixed graphical and textual DSLs in a single pipeline. Therefore, we consider requirements RQ-1, RQ-2, and RQ-3 satisfied, since non-domain experts successfully managed to produce working and deployable systems in under an hour.

Apart from the crisp data evaluation, of utmost importance is also the user-oriented acceptance of the methodology/system, which can be captured via the System Usability Scale. The SUS can measure user experience and the usability of using the proposed methodology via the LocSys platform, consisting of 10 questions that yield a single number (in the range of 0–100) mapped into a set of SUS grades (A+ to F) representing a unified measure of system usability performance in terms of effectiveness, efficiency, and overall ease of use. The SUS distribution is visualized in Figure 9, and the respected analyses are presented in Table 1 and Table 2, where the grades, ratings, acceptance rate [43], and NPS (net promoter score) categories associated with raw SUS scores [42] are also included. The SUS results indicate that the participants assigned a high usability rating to our approach, as 61.91% of them were classified in the A+, A, and A− grades. This outcome is backed up by the average SUS score, which is 80.65 out of 100, representing a grade of A−, which is considered Excellent. Based on this feedback, it can be inferred that RQ-3 has been satisfied, i.e., the interface via which the proposed methodology was implemented is simple and intuitive.

### 5.3. Statistical Analysis

In order to validate the SUS results, univariate analysis of variance (ANOVA) was used for investigating the effect of programming, Python, CPS, smart automations, and DSLs experience/level on the SUS score, by using the experience characteristics as independent variables and the SUS score as the dependent variable. Levene’s test was used to assess the equality of variances and Box’s M Test for the assessment of the equivalence of covariance matrices. Partial eta-squared ($\eta^{2}$) was used for the estimation of the effect size.

According to the results presented in Table 3, there was no significant effects of the participants’ experience in any domain on the SUS score F(5,45)=1.33, *p* < .05. The results indicate that the participants found the LocSys approach highly usable, regardless of their technical experience in the programming, smart automations, and CPS domains. This fact allows us to state that the multi-DSL approach is suitable for citizen developers and even non-technical CPS/IoT enthusiasts that desire to control such systems, satisfying RQ-2 and RQ-3.

### 5.4. Limitations and Threats to Validity

Empirical software engineering (ESE) research focuses on the application of empirical methods at any phase of the software development life cycle [44]. The three most common methods of empirical research are surveys, case studies, and experiments. Surveys are conducted through questionnaires or interviews, while case studies examine phenomena in a real-world context. Experiments, on the other hand, have a limited scope and are typically conducted in a laboratory setting with a high level of control. Our case implemented an amalgamation of a survey and a case study, since real-world data were captured from the users’ interaction with LocSys, and the users provided responses to a questionnaire.

To this extent, the number of involved participants is considered a threat to validity, since even though 84 is an adequate number for the data analysis, only 45 participants answered the questionnaire. Furthermore, it could be argued that the LocSys approach in the empirical evaluation may pose a threat to validity, as the relevant SUS results and usability and productivity measurements are not compared to other low-code development platforms for smart environments. This is due to the fact that bibliography does not include scientific evaluations of such solutions based on usability and productivity measurements for citizen developers. We intend to extend our research and conduct dedicated workshops to compare productivity and usability metrics against traditional development methodologies and tools for end-to-end development of smart environments, where applicable, such as HomeAssistant environment definition and automation engine tools. Although, it should be noted that LocSys application deployments (such as automations and virtual entities) have direct integration with HomeAssistant functionality, by pointing to a single MQTT broker for the communication layer of both platforms. In this sense, we argue that our approach can be used next to state-of-play solutions for smart environments to boost usability and productivity.

## 6. Conclusions and Future Work

In this work, we evaluate our approach of developing applications for smart environments using multiple high-level DSLs against usability, user acceptance, and productivity metrics. Specifically, we are introducing the decomposed multi-DSL methodology, where complex domains are divided into layers and aspects, and DSLs for each layer/aspect are created, allowing for separation of concerns, as experts in different domains can contribute the most by focusing on the most relevant DSL. These DSLs share common core meta-models and grammars, making transformations between languages easier to perform, and also allowing for the creation of DSL pipelines. To materialize this approach, the LocSys platform was developed, which allows for integrating DSLs and transformations, while it also provides a mechanism for defining pipelines (languages and transformations between languages) for complex domains where a multi-DSL approach is used. A key aspect of the methodology, as stated in Section 3.1, is the separation of the application from other parts of a smart environment, assuming already connected devices (entities in general) to a message broker. Thus, DSLs use high-level concepts to describe application-level aspects, such as automation processes and monitoring dashboards.

The results of the evaluation of the approach were validated by the utilization of four heterogeneous DSLs, spanning into two out of the three CPS layers, as defined in Section 3. These indicated that the 86 participants, that can be described as non-experts of the CPS domain, successfully developed a CPS for a smart environment, by defining a blueprint, populating it with smart devices, composing automations, and building monitoring dashboards, at a rate of 96.42%. Furthermore, the decomposed multi-DSL approach, materialized within LocSys platform, was well accepted by the participants in terms of usability, as indicated by the SUS scores, achieving an average score of 80.65 out of 100, representing a grade of A- (Excellent).

The challenges of the proposed multi-DSL methodology are twofold. (a) The most time-consuming part is the development of the DSLs from the domain expert. Once the DSLs are concluded, they provide a high level of abstraction using domain-specific concepts and, thus, offer a great acceleration of the end users’ performance. (b) The acceptance of the abstraction layer and language semantics, by end users, is key to improving usability and performance metrics. Thus, a DSL must have clear and human-friendly grammar (concrete syntax). Regarding the utilization of our approach in real-world scenarios, it is considered a case of the SmAuto and Codin code generators to build more robust platform-specific software. Our approach requires no changes for connected devices because the application layer communicates via a unified communication layer (e.g., message brokers such as MQTT and AMQP). Generated software is not concerned with the nature of the underlying entities, physical or virtual. Of course, one should argue that this is a limitation of our approach. However, we strongly believe that quality and security metrics should be considered at design time (DSL models) and runtime (generated software) for real-world applications in smart environments.

Future work includes the full evaluation of the decomposed multi-DSL approach, by experimenting on all three layers of CPS, implementing DSLs at the first layer (device). For example, DSLs that integrate robots or other complex devices can be created, sub-domains that are part of CPS but have their own inherent peculiarities. This way, the methodology will be criticized against the realistic task of integrating real devices into the pipeline, building automation processes, and monitoring dashboards.

## Figures and Tables

**Figure 1 sensors-25-03951-f001:**
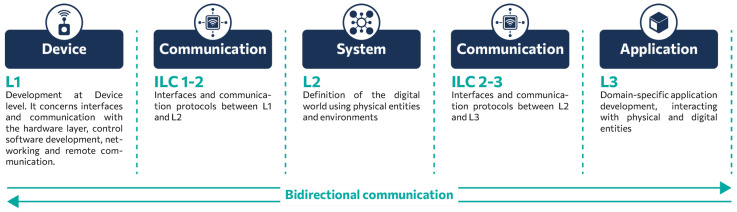
Three-layered architecture of a CPS.

**Figure 2 sensors-25-03951-f002:**
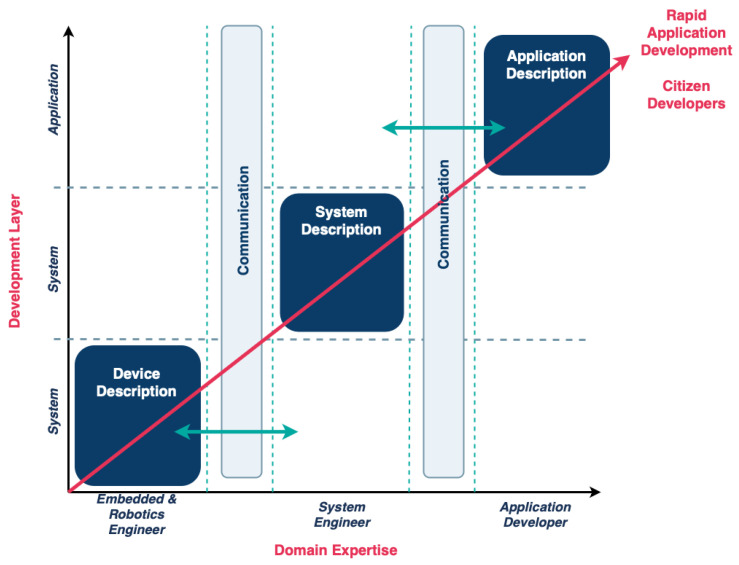
Domain decomposition based on development layer and domain expertise of DSL users. Green arrows denote communication between software modules belonging in adjacent layers.

**Figure 3 sensors-25-03951-f003:**
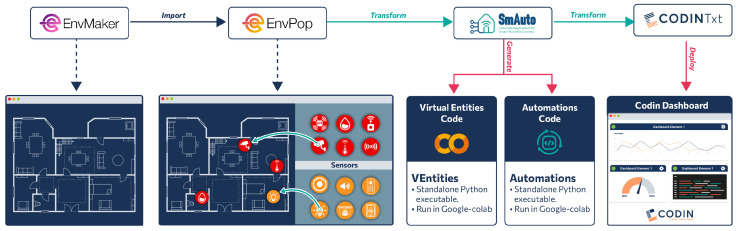
The selected cyber-physical systems pipeline that was implemented as showcased.

**Figure 4 sensors-25-03951-f004:**
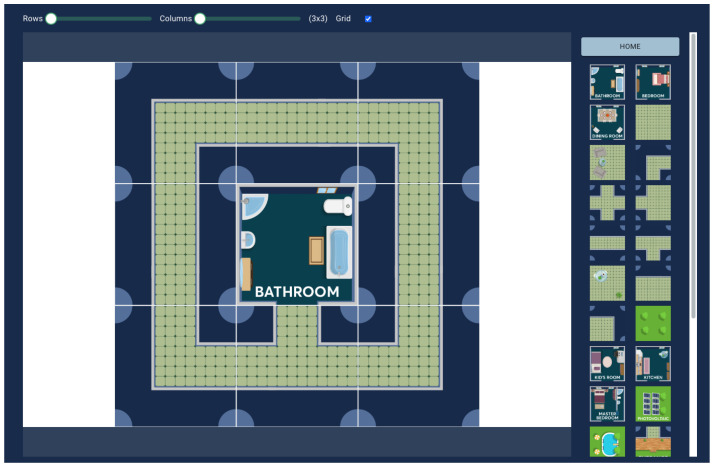
Example of a 3 × 3 EnvMaker model.

**Figure 5 sensors-25-03951-f005:**
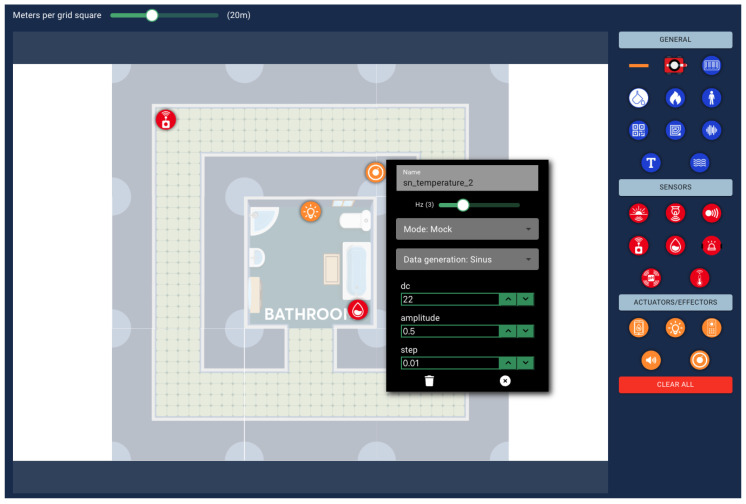
EnvPop model example, using the created EnvMaker model.

**Figure 6 sensors-25-03951-f006:**
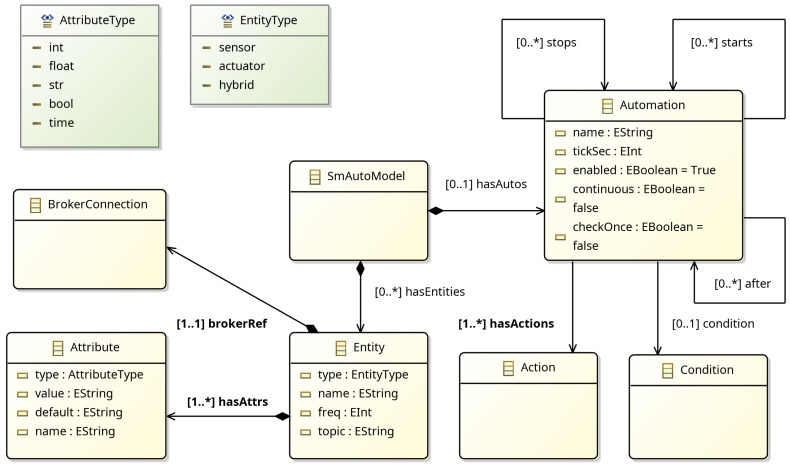
SmAuto meta-model.

**Figure 7 sensors-25-03951-f007:**
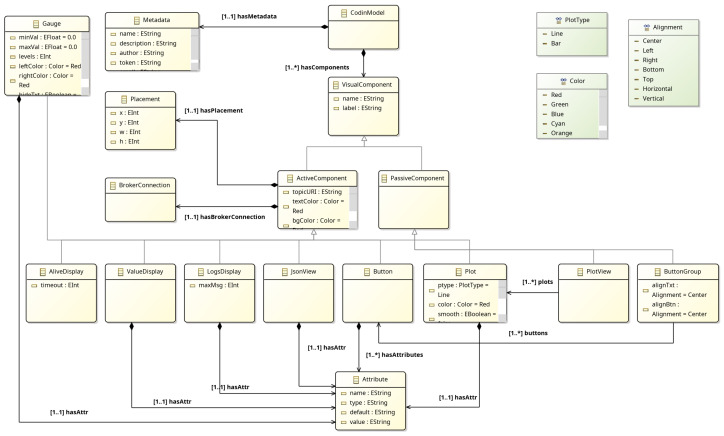
CodinTxt meta-model.

**Figure 8 sensors-25-03951-f008:**
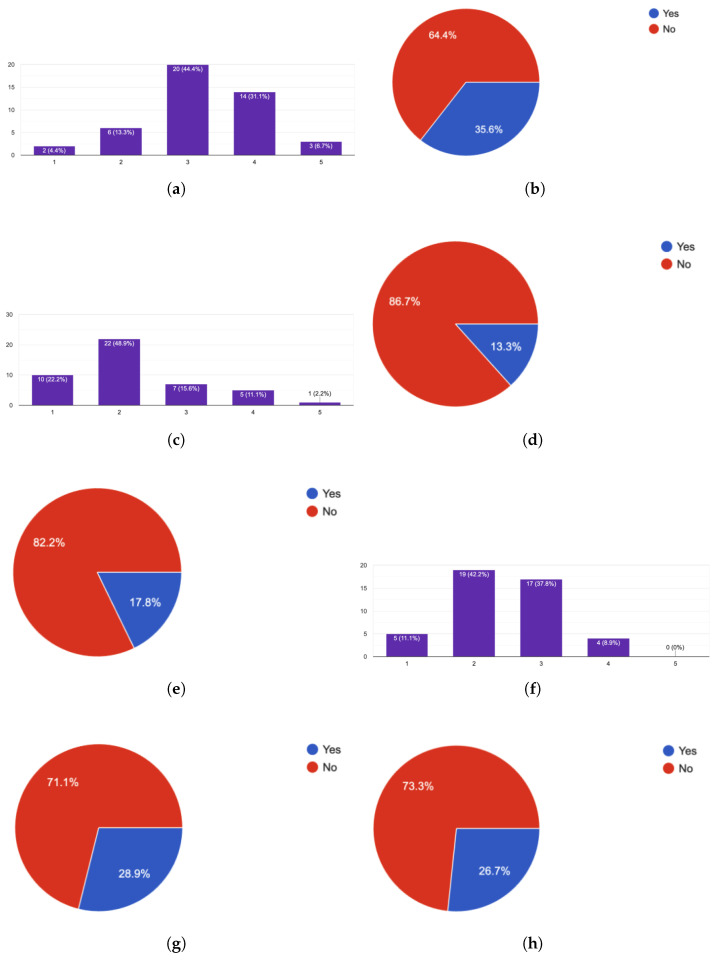
Evaluation responses regarding the background (knowledge and experience) of the participants. (**a**) Question: What is your level of knowledge in software programming? (**b**) Question: Have you even used a DSL for software development? (**c**) Question: What is your level of knowledge in the field of modern CPSs? (**d**) Question: Have you been involved in any way in the development of a CPS? (**e**) Question: Have you been involved in any way in the development of a cyber-physical application? (**f**) Question: What is your level of knowledge in the field of smart infrastructure and process automation in smart homes? (**g**) Question: Have you been involved in any way in the development of a smart home system? (**h**) Question: Have you been involved in any way in the development of an application for smart homes (or smart infrastructure in general)?

**Figure 9 sensors-25-03951-f009:**
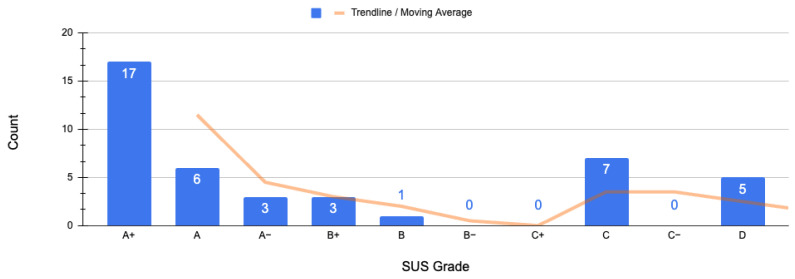
SUS grades distribution.

**Table 1 sensors-25-03951-t001:** Analysis of the overall percentage for each grade of the SUS scale.

SUS Grade	A+	A	A−	B+	B	B−	C+	C	C−	D
Count	17	6	3	3	1	0	0	7	0	5
Percentage (%)	40.48	14.29	7.14	7.14	2.38	0.00	0.00	16.67	0.00	11.90

**Table 2 sensors-25-03951-t002:** SUS results including average, minimum, and maximum values.

Metric	SUS Score	Grade	Adjective	Acceptability	NPS
Average	80.65	A−	Excellent	Acceptable	Promoter
Minimum	57.5	D	-	Marginal	Passive
Maximum	100	A+	Best Imaginable	Acceptable	Promoter
Median	78.75	B+	Good	Acceptable	Promoter

**Table 3 sensors-25-03951-t003:** Effect of programming, Python, CPS, automations, and DSLs experience/level on the SUS score, according to ANOVA analysis.

SUS Score	df	F	*p*	$\eta^{2}$
Programming level	15	1.050	.441	.377
Python level	15	1.622	.135	.483
CPS level	15	1.539	.163	.470
Automations level	15	1.521	.169	.467
DSLs experience	15	0.693	.758	.286

## Data Availability

Dataset available on request from the authors.

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
