# Peer review of "LocSys: A Low-Code Paradigm for the Development of Cyber-Physical Applications"

_sensors, 2025, doi:10.3390/s25133951_

Round 1
Reviewer 1 Report
Comments and Suggestions for Authors
The contributions of this work are well presented, and the overall presentation is commendable. While the reported results are promising, they are not reproducible based on the information currently provided. To improve reproducibility, the authors are strongly encouraged to include a detailed description of the experimental setup and, ideally, share the source code through a platform such as GitHub.
A more in-depth discussion of the results is also recommended—particularly to highlight the factors contributing to the observed performance improvements over existing methods. Additionally, the authors should elaborate on any challenges encountered during experimentation, as well as the potential implications or limitations if the proposed approach were to be implemented in real-world applications.
The literature review includes several outdated references. It is strongly recommended that the authors revise this section to incorporate more recent and relevant works. For example, recent developments involving technologies such as WebAssembly (WASM) have not been adequately considered.
Lastly, providing a clear and concise summary of both the advantages and limitations of the proposed method would further strengthen the manuscript.
Author Response
Thank you very much for your time and thoughtful review of our manuscript. We truly appreciate your valuable comments and suggestions. Please find below our detailed responses to each point raised. The corresponding revisions have been implemented in the manuscript and are clearly indicated by specific line numbers for your convenience. We believe these improvements have strengthened the quality and clarity of the paper.
Comment 1.1: The contributions of this work are well presented, and the overall presentation is commendable. While the reported results are promising, they are not reproducible based on the information currently provided. To improve reproducibility, the authors are strongly encouraged to include a detailed description of the experimental setup and, ideally, share the source code through a platform such as GitHub.
Response 1.1: Thank you for your comment. The usability results presented in the current manuscript, as mentioned in section 5 (Evaluation), are based on the SUS questionnaire that was handed to the participants, and the experimental setup of the workshop is also described in detail in section 5 (Evaluation). Even though sharing the source code of the LocSys platform would be ideal for reproducibility, it is currently closed source, since it constitutes a key research asset of our team. Nevertheless, its use is free, thus anyone can create an account and reproduce the results by creating DSL models, or even host workshops like our own.
Comment 1.2: A more in-depth discussion of the results is also recommended—particularly to highlight the factors contributing to the observed performance improvements over existing methods. Additionally, the authors should elaborate on any challenges encountered during experimentation, as well as the potential implications or limitations if the proposed approach were to be implemented in real-world applications.
Response 1.2: Thank you for your useful comment. Based on this, we have made changes in the “Conclusions and Future Work” section and included a paragraph stressing out the challenges and limitations of the proposed method, beyond others (lines 817-831). We have also included discussion on real-world applications and the importance of the decomposition into layers, as well as the importance of quality and security metrics of the generated software (lines 793-807). As discussed in the manuscript, the application layer is connected to other parts of the system via a unified communication layer implemented using message broker technologies and the commonly used MQTT and AMQP protocols. Generated software is not concerned with the nature of the underlying entities, physical or virtual.
Comment 1.3: The literature review includes several outdated references. It is strongly recommended that the authors revise this section to incorporate more recent and relevant works. For example, recent developments involving technologies such as WebAssembly (WASM) have not been adequately considered.
Response 1.3:
We acknowledge the reviewer’s comment; thus, we have enriched the literature review with more recent references. Regarding the proposed WebAssembly reference, after research, we found out that it was designed to permit near-native code execution speed in the web browser. In our case, the proposed DSLs do not execute code or models in the browser, but they simply allow end users to “develop” models via an integrated web environment. Operations such as validation, generation, deployment, and transformations are offered from the platform, and the generated software is GPL source code (e.g., Python and JavaScript). This generated software can be downloaded on demand (for specific DSL supporting M2Ts) and can be executed offline, on the end user’s resources. Therefore, we believe that WASM is unrelated to our work, and that is why we did not include it in the state-of-the-art section.
Comment 1.4: The introduction contains some explanations more related to background issues than to introductory aspects of motivation and justification. This information should be moved to another section.
Response 1.4: We have revised the manuscript and performed changes in the Introduction section, such as removing excessive background information, and focusing on the motivation, the under-study research questions, and the contribution of our work. We believe that now the structure of the paper has been improved; thank you for your comment.
Comment 1.5: Lastly, providing a clear and concise summary of both the advantages and limitations of the proposed method would further strengthen the manuscript.
Response 1.5: Indeed, we agree with the reviewer. Therefore, we made changes in the “Conclusions and Future Work” section and included a paragraph stressing out the challenges and limitations of the proposed method. Furthermore, changes were performed to stress out the limitations and the advantages of the proposed multi-DSL approach, assuming already connected devices.
Reviewer 2 Report
Comments and Suggestions for Authors
The manuscript presents an interesting and promising development in software engineering area. Nevertheless, major revision is needed to properly explain research aspects of this R&D study.
- No research problem is defined. The LocSys platform is a technical result. What important research results do support the platform (make the platform workable, correct and efficient)? How do the results contribute to the body of scientific knowledge? Appropriate updates are needed in all sections of the manuscript, including Abstract.
- The presented software engineering models need more mathematical support. Please, augment the models with parameters. Such parameters are subject to be used and evaluated in experiments.
- The experimental study looks as interesting experimental research. Nevertheless, deeper insight to the data is needed to make the conclusions (i) properly grounded and (ii) useful for other experts in the area.
Comments on the Quality of English LanguageNot an expert in English language.
Scientific writing style (in English) can be improved.
Author Response
Thank you very much for your time and thoughtful review of our manuscript. We truly appreciate your valuable comments and suggestions. Please find below our detailed responses to each point raised. The corresponding revisions have been implemented in the manuscript and are clearly indicated by specific line numbers for your convenience. We believe these improvements have strengthened the quality and clarity of the paper.
Comment 2.1: No research problem is defined. The LocSys platform is a technical result. What important research results do support the platform (make the platform workable, correct and efficient)? How do the results contribute to the body of scientific knowledge? Appropriate updates are needed in all sections of the manuscript, including Abstract.
Response 2.1: Indeed, the lack of clear motivation was a drawback of the paper, identified by the other reviewers as well. To resolve this, we have revised the manuscript, including the abstract, to highlight the key aspects, background motivation, the main contributions, and research outcomes of our study. More specifically, we stress the utilization of the SUS scale to evaluate the usability and acceptance of our methodology by non-experts, such as the case of citizen developers. These alterations can be seen in red color, all over the resubmitted manuscript.
Comment 2.2: The presented software engineering models need more mathematical support. Please, augment the models with parameters. Such parameters are subject to be used and evaluated in experiments.
Response 2.2: The contributed meta-models of the current study refer to the CodinTxt DSL, which does not have a mathematical background, as it is a mapping of entities to dashboard elements (e.g., to map a temperature sensor to a gauge or a plot component). Regarding SmAuto meta-models, these are contributed to a previously published manuscript (https://doi.org/10.1016/j.pmcj.2024.101931), as described in section 4. In case the comment's implications were not fully comprehended by our side, further elaboration and counsel would be appreciated.
Comment 2.3: The experimental study looks as interesting experimental research. Nevertheless, deeper insight to the data is needed to make the conclusions (i) properly grounded and (ii) useful for other experts in the area.
Response 2.3: Even though the comment tries to improve the understandability of the paper, it is too general for us to understand what the reviewer had in mind. After including an ANOVA analysis on the results, in order to validate them (as proposed by reviewer #3), we believe the experimentation analysis is complete from our side. If you have a suggestion that could improve our results, further elaboration and counsel would be appreciated.
Reviewer 3 Report
Comments and Suggestions for Authors
The paper can be enhanced by addressing the following:
The abstract needs significant revision to reflect what the study is about, rewriting as some sentences are not smooth and hard to follow, such as “...may be discouraging for technical teams to initiate such endeavors, let alone solo developers…” Also, the abstract is written in passive voice; the authors should write this in the present tense. In addition, it should include some of the study outcomes in a quantifiable manner.
The introduction is insufficient and needs much improvement. Terms like “model-driven operators” are introduced without enough context.
Lack of thesis or motivation: Why is this survey needed now? What gap does it fill in existing literature?
A key weakness of the paper lies in its methodology section, which is missing to validate the study's rigor. The methodology section is a key section and is one of the critical parts of a study. This raises concerns about potential bias or incompleteness in the study. The authors should include a separate section called methodology.
Some performance evaluation metrics are missing; the authors should include some of the following in the revised version of the paper: error rate during modeling, learning curve (e.g., first-time success, reattempts), developer satisfaction (survey-based), if possible.
A table of acronyms would enhance the readability of the paper.
The authors should conduct more statistical analysis, particularly ANOVA to validate the results and provide more credibility of the results.
Comments on the Quality of English Language
The paper needs some editing and proofreading.
Author Response
Thank you very much for your time and thoughtful review of our manuscript. We truly appreciate your valuable comments and suggestions. Please find below our detailed responses to each point raised. The corresponding revisions have been implemented in the manuscript and are clearly indicated by specific line numbers for your convenience. We believe these improvements have strengthened the quality and clarity of the paper.
Comment 3.1: The abstract needs significant revision to reflect what the study is about, rewriting as some sentences are not smooth and hard to follow, such as “...may be discouraging for technical teams to initiate such endeavors, let alone solo developers…” Also, the abstract is written in passive voice; the authors should write this in the present tense. In addition, it should include some of the study outcomes in a quantifiable manner.
Response 3.1: Indeed, the lack of clear motivation was a drawback of the paper, identified by the other reviewers as well. To resolve this, we have revised the abstract of the manuscript as proposed by the reviewer to reflect the key aspects of our study, the main contribution, and research outcomes. Regarding the passive voice we write all our publications this way, as it is our preferred style; we hope that the reviewer understands.
Comment 3.2: The introduction is insufficient and needs much improvement. Terms like “model-driven operators” are introduced without enough context.
Response 3.2: Thank you for this comment, as it aligns with the relevant comments of other reviewers. We have revised the manuscript and performed changes in the Introduction section, such as removing excessive background information, and focusing on the motivation, the under-study research questions, and the contribution of our work.
Comment 3.3: Lack of thesis or motivation: Why is this survey needed now? What gap does it fill in existing literature?
Response 3.3: This comment overlaps with the two previous comments of the reviewer, thus we believe that the changes described in the previous answers cover that too.
Comment 3.4: A key weakness of the paper lies in its methodology section, which is missing to validate the study's rigor. The methodology section is a key section and is one of the critical parts of a study. This raises concerns about potential bias or incompleteness in the study. The authors should include a separate section called methodology.
Response 3.4: Thank you for your comment. Based on it, we introduced a new section called Methodology (section 3), which describes in a more theoretical way out approach. This approach is instantiated via the LocSys platform, now described in section 4. We believe the current structure is much better than before, thus we thank the reviewer.
Comment 3.5: Some performance evaluation metrics are missing; the authors should include some of the following in the revised version of the paper: error rate during modeling, learning curve (e.g., first-time success, reattempts), developer satisfaction (survey-based), if possible.
Response 3.5: Although the reviewer rightly points out some evaluation metrics, unfortunately, the platform did not support the metrics related to error rate during modeling and learning curve. We are currently investigating the use of these metrics for the continuation of the current study by integrating a behavior-capturing and storing mechanism in the LocSys platform. This way, such metrics can be retrieved and post-analyzed.
Comment 3.6: A table of acronyms would enhance the readability of the paper.
Response 3.6: Abbreviations are already included in the manuscript based on the MDPI template for the Sensors Journal. We have revised the abbreviations section and included more acronyms. Thank you for pointing that out.
Comment 3.7: The authors should conduct more statistical analysis, particularly ANOVA to validate the results and provide more credibility of the results.
Response 3.7: Based on this comment, we have performed an ANOVA analysis (section 5.3) to investigate whether the various experience levels (CPS, Programming, Python etc) of the participants affected the SUS score. The results indicate that the SUS score is not affected by the experience levels (F(5, 45) = 1.33, p < .05), thus the participants found the approach usable, regardless of their technical background. We believe this addition has strengthened our work, thus we thank the reviewer again.
Round 2
Reviewer 3 Report
Comments and Suggestions for Authors
All comments have been addressed.